# JUMPY RECURRENT NEURAL NETWORKS

## ABSTRACT

Recurrent neural networks (RNNs) can learn complex, long-range structure in time series data simply by predicting one point at a time. Because of this ability, they have enjoyed widespread adoption in commercial and academic contexts. Yet RNNs have a fundamental limitation: they represent time as a series of discrete, uniform time steps. As a result, they force a tradeoff between temporal resolution and the computational expense of predicting far into the future. To resolve this tension, we propose a Jumpy RNN model which does not predict state transitions over uniform intervals of time. Instead, it predicts a sequence of linear dynamics functions in latent space and intervals of time over which their predictions can be expected to be accurate. This structure enables our model to jump over long time intervals while retaining the ability to produce fine-grained or continuous-time predictions when necessary. In simple physics simulations, our model can skip over long spans of predictable motion and focus on key events such as collisions between two balls. On a set of physics tasks including coordinate and pixel observations of a small-scale billiards environment, our model matches the performance of a baseline RNN while using a fifth of the compute. On a real-world weather forecasting dataset, it makes more accurate predictions while using fewer sampling steps. When used for model-based planning, our method matches a baseline RNN while using half the compute.

## 1 INTRODUCTION

It is said that change happens slowly and then all at once. Billiards balls move across a table before colliding and changing trajectories; water molecules cool slowly and then undergo a rapid phase transition into ice; and economic systems enjoy periods of stability interspersed with abrupt market downturns. That is to say, many time series exhibit periods of relatively homogeneous change divided by important events. Despite this, recurrent neural networks (RNNs), popular for time series modeling, treat time in uniform intervals – potentially wasting prediction resources on long intervals of relatively constant change.

One reason for this is that standard RNNs are *sequence* models without an explicit notion of time. Instead, the amount of time represented by a single RNN update is implicitly set by the training data. For example, a model trained on sequences of daily average temperatures has an implicit time step of a day. For a fixed computational budget, this introduces a trade-off between fidelity and temporal range. A model trained at a resolution of one time step per minute would require over 10K iterations to make a prediction for one week in the future. At the other end of the spectrum, a one-week resolution model could achieve this in a single step but could not provide information about the intervening days. As such, selecting a point on this spectrum is a troublesome design decision.

In this work, we present Jumpy RNNs, a simple recurrent architecture that takes update steps at variable, data-dependent time-scales while being able to provide dense predictions at intervening points. The core innovation is to define the hidden state as a continuous, piece-wise linear function of time. Specifically, each Jumpy RNN step predicts not only a hidden state $h_i$, but also a hidden *velocity* $\dot{h}_i$ and a span of time $\Delta$ over which the linear latent dynamics $h(t) = h_i + \dot{h}_i(t - i)$ should be applied. Our model then jumps forward in time by $\Delta$ before updating again. Any intermediate time step can be produced by decoding the corresponding hidden state $h(t)$.

During training, our model learns to use these functions to span the non-uniform time durations between key events, where key events emerge as time points where linear latent extrapolation is

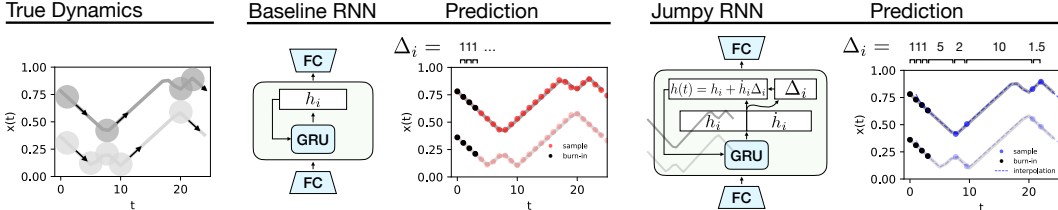

Figure 1: Predicting the dynamics of two billiards balls (left) using a baseline RNN cell (center) and a Jumpy RNN cell (right). Whereas the baseline model produces a hidden state $h_t$ at each time step, our jumpy model predicts a continuous-time hidden state, over a predicted interval $\Delta_i$. This allows it to skip over long spans of predictable motion and focus on key events such as collisions.

ineffective. In Figure 1, for example, we see that our model updates at the collision points between the two balls and the walls. During time spans when the balls are undergoing constant motion, our model does not perform cell updates. In contrast, a standard RNN must tick uniformly through time.

We demonstrate our proposed model in several physical dynamics prediction tasks. We show Jumpy RNNs achieve comparable performance to the baseline while being between three and twenty times more efficient to sample. This includes settings with non-linear pixel-based observations. Further, we show that our model outperforms RNNs with any fixed step length, showing the importance of data-dependent step sizes. Finally, we demonstrate that a learned Jumpy RNN dynamics model can be leveraged as an efficient forward predictor in a planning domain. Our key contributions are to:

– Identify a trade-off between temporal resolution and the computational expense of RNNs,
– Propose Jumpy RNNs, which make jumpy predictions and interpolate between them,
– Show empirically that Jumpy RNNs are efficient and effective at jumpy time series prediction.

## 2 JUMPY RECURRENT NEURAL NETWORKS

Consider a continuous-time function $x(t)$ sampled at uniform time steps to form the sequence $x_0, x_1, \ldots, x_T$. We study the problem of generative modeling, where given an initial set of observations, the goal is to auto-regressively predict a likely continuation.

**Standard RNN.** The per-step RNN computation during auto-regressive sequence generation is:

$$h_t = \text{RNNCell}(\ h_{t-1},\ \phi(\hat{x}_t)\ ), \quad \hat{x}_{t+1} = f(h_t)$$

where $\phi(\cdot)$ and $f(\cdot)$ are observation encoders and decoders respectively, and $h_t$ is the hidden state. This computation is performed at each time-step even when the dynamics of $x(t)$ are simple (or even constant) for long stretches of time. One way to linearly reduce the computation required to predict into the future is to increase the time-span $\Delta$ between RNN ticks. Standard RNNs, however, then lose the ability to predict at clock times in between the $\Delta$ time steps. This introduces a trade-off between predictive resolution and the computational cost of predicting far into the future.

### 2.1 JUMPY RNN ARCHITECTURE

**Continuous Hidden Dynamics with Constant Jumps.** Our first step toward resolving the trade-off is to upgrade the standard RNN so that it can learn to linearly interpolate a continuous-time hidden state $h(t)$ between updates. Let $\Delta$ be the time between RNN ticks such that RNN tick $i$ (starting at $i = 0$) corresponds to continuous time point $\tau_i = i\Delta$. For update $i$, the RNN predicts both a hidden state $h_i$ and hidden *velocity* $\dot{h}_i$ that describes how the hidden state $h(t)$ evolves over the time interval $[\tau_i, \tau_i + \Delta]$. Specifically, the operation of this linear-dynamics RNN with constant jump is given by:

$$h_i, \dot{h}_i \quad = \quad \text{RNNCell}\left(\ \left[h_{i-1},\ \dot{h}_{i-1}\Delta\right],\ \phi\left(\hat{x}(\tau_i)\right)\ \right) \tag{1}$$

$$h(t) \quad = \quad h_i + (t - \tau_i)\dot{h}_i \quad \text{for}\quad t \in [\tau_i, \tau_i + \Delta] \tag{2}$$

$$\hat{x}(t) \quad = \quad f(h(t)) \tag{3}$$

where $[\cdot, \cdot]$ denotes concatenation and $\hat{x}(t)$ is the continuous time output prediction that can be immediately produced on demand for any time in between ticks. Under this model, the hidden state

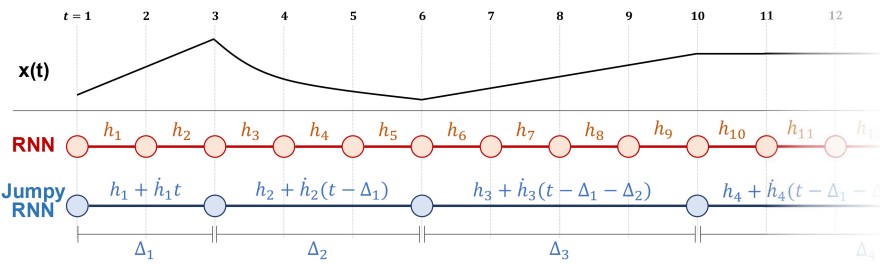

Figure 2: Our Jumpy RNN defines a continuous-time, piece-wise linear hidden state.

evolves as a piece-wise linear function of time, with transitions between linear functions occurring at RNN updates. This linearity constraint provides a strong and natural inductive bias and allows for fast interpolation. Importantly, the hidden-state linearity does not translate to linearity in output space, as the learned decoder $f(\cdot)$ can be an arbitrarily complex non-linear function.

While the above model can interpolate between time steps, it is still constrained by a constant jump width $\Delta$. Manually selecting $\Delta$ is difficult since the ideal value depends on the complexity of $x(t)$ and the encoder and decoder capacity. Further, $x(t)$ may vary in complexity over time such that a single jump width is insufficient.

**Jumpy RNNs.** The Jumpy RNN upgrades the above model to dynamically adjust $\Delta$ throughout the sequence as shown in Figure 2. Specifically, we now predict a duration $\Delta_i$ in addition to linear latent dynamics $\left[h_i, \dot{h}_i\right]$ at each RNN step. This duration encodes the time span over which the corresponding linear latent dynamics accurately approximate $x(t)$. RNN ticks now occur at variable points in time so that RNN step $i$ occurs at time point $\tau_i = \sum_{j=0}^{i-1} \Delta_j$. Now our update is:

$$h_i, \dot{h}_i = \text{RNNCell}\left(\ \left[h_{i-1},\ \dot{h}_{i-1}\Delta_{i-1}\right],\ \phi\left(\hat{x}(\tau_i)\right)\ \right) \tag{4}$$

$$\Delta_i = \text{LeakyReLU}(Wh_i + b) + 1 \tag{5}$$

$$h(t) = h_i + (t - \tau_i)\dot{h}_i \quad \text{for} \quad t \in [\tau_i, \tau_i + \Delta_i] \tag{6}$$

where again $\hat{x}(t) = f(h(t))$. In practice, this approach can be implemented on top of *any* existing RNN variant, for example, GRUs. Training the jumpy model, however, introduces new complications since the optimal dynamic steps $\Delta_i^*$ are unknown and interact with the rest of the hidden state. This is partly addressed by the inductive bias of Eq. 5, which encourages our model to always jump forward in time by at least one time step. The next section details our approach to effective training.

## 2.2 TRAINING JUMPY RNNS

Even though our model is defined over continuous time, the training data for time-series models typically consists of observation sequences of the form $x_0, x_1, x_2, \ldots, x_T$. For the remainder we will assume that there is one time unit between each sample, i.e. $x_t = x(t)$. We now describe how to train Jumpy RNNs in this setting by constructing a loss function specifically suited to our jumpy model.

**Supervising Predictions.** Supervising for prediction accuracy simply requires evaluating the model at each discrete time step and applying the loss: $\mathcal{L}_x = \sum_{t=0}^{T} \ell(x_t, \hat{x}(t))$, where $\ell(\cdot, \cdot)$ is a loss between predicted and actual values. In concept, this loss alone may be sufficient for optimizing accuracy, however, it does not encourage the model to take large jumps (predict large $\Delta_i$) when possible. In fact, it may encourage small jumps as an easier optimization path. Thus, below we augment the loss to encourage jumpiness.

**Supervising Dynamic Jumps.** We interpret the the jump width $\Delta_i$ as the duration that the linear dynamics can accurately approximate x(t). To formalize this, we say a training point $x_t$ is accurately approximated if $\ell(x_t, \hat{x}(t)) < \epsilon$, where $\epsilon$ is a hyperparameter. Ideally, we would like the Jumpy RNN to produce the maximum jump width that maintains this constraint.

More formally, given fixed network weights the optimal jump width $\Delta_i^*$ at step $i$ is the solution to:

$$\max_{\Delta_i \geq 1} \Delta_i \quad \text{s.t.} \quad \ell(x_t, \hat{x}(t)) < \epsilon \quad \forall \{x_t \mid t \in [\tau_i, \tau_i + \Delta_i]\}, \tag{7}$$

which seeks the largest jump that does not violate the loss threshold for any intervening observation. This can be solved using a simple forward line search that starts at $\tau_i$ and moves forward through the samples until our model's prediction loss under the current dynamics exceeds $\epsilon$. If this occurs on the first step (implying no time passes between ticks), we set $\Delta_i^*$ to 1. In Appendix A.1 we show this can be efficiently vectorized for an entire batch as part of a single pass through the sequence.

We compute the optimal $\Delta_i^*$ on-the-fly during training and jump forward accordingly – effectively ignoring the predicted $\Delta_i$ as in standard teacher-forcing style training. To update the model to predict the correct $\Delta_i$, we augment the prediction loss $L_x$ with a jump size loss $\mathcal{L}_\Delta = \sum_i ||\Delta_i - \Delta_i^*||_2^2$. At inference, the predicted $\Delta_i$ is used to determine jump length.

**Interpreting Error Threshold $\epsilon$.** The error threshold $\epsilon$ is a key hyperparameter introduced in our approach. Examining Equation 7, $\epsilon$ acts as a trade-off between jumpiness and approximation error. In the extreme of setting $\epsilon = 0$, the model predictions will likely never be below $\epsilon$, $\Delta_i$ will remain at 1, and our model will reduce back to a standard RNN that updates at each time step. At the other extreme, choosing $\epsilon = \infty$ forces our model to predict the entire time series with a single linear latent transition. In practice, we found that setting $\epsilon$ to the final training loss of the baseline model yields 3-10x jumpy models with the same test error as the baseline model.

## 3 EXPERIMENTS

In this section, we evaluate our model on a suite of tasks including both coordinate and video representations of a small-scale billiards simulation. Our model attains test errors and autoregressive sample errors comparable to that of the baseline while doing so in 3-20x fewer steps. Lastly, we show how our model can be leveraged for more efficient model-based planning.

**Implementation Details.** We use the same architecture and training regime across all experiments unless otherwise specified. We use a GRU cell with 128 hidden units and our encoder and decoder networks are four-layer residual MLPs with 128 hidden units and ReLU activations. We train our model with ADAM for $10,000$ steps with a batch size of 256, a learning rate of $1 \times 10^{-3}$ and a decay schedule of $\gamma = 0.9$ every 1000 steps. We used a mean square error loss function to train both teacher forcing predictions and $\Delta$ predictions, and scaled the latter by $1 \times 10^{-5}$ relative to the former. For model selection, we used early stopping.

**Jump Bootstrapping.** The initial transition to jumping more than one step is challenging in that it requires latent dynamics to extrapolate essentially without supervision. To bootstrap the process, we force a jump even when error is greater than $\epsilon$ in 1% of cases. This leads to more stable results; for more discussion see Appendix A.2.

### 3.1 MODELING CONSTANT MOTION

To begin, we consider simple settings that describe constant motion of a moving object. These settings demonstrate the limitations of standard RNNs, our model's ability to translate linear latent dynamics to non-linear motion, and the plotting strategies we will use for visualizing all models.

**Lines.** First, we consider the simple task of modeling linear motion. We generate 10,000 training trajectories corresponding to linear motion along one dimension in $(x, y)$ Cartesian coordinates. Specifically, $\{(x_0, y_0), \ldots, (x_{20}, y_20)\}$ where $x_t = t$ and $y_t = c$ where $c$ is a random constant in $(0, 1)$. Plotted out, this corresponds to a flat, straight line starting at a random height. Unsurprisingly, our Jumpy RNN model quickly learns to jump to the end of the sequence (mean $\Delta_i$ of 20) while the standard RNN operates at a fixed rate. We trained on 90% of the data and evaluated on the remainder and both models trivially achieved very low error ($10^{-8}$). See the Appendix A.4 for samples.

**Circles.** Our second task also involves modeling constant motion; but in this case in terms of constant tangential velocity while moving in a circle. While highly non-linear in the Cartesian observation space, the motion *is* linear in polar coordinates. This experiment examines our model's ability to encode non-linear observations with linear latent dynamics – effectively learning a Cartesian-to-Polar conversion. As before, our dataset contained $10,000$ examples, this time with 25 time steps each. Each sequence started from a random angle in $(0, 2\pi)$ and a random radius in $(1, 2)$. We fixed the tangential velocity to a constant value across all trajectories based on the intuition of a particle undergoing constant velocity along 1D manifolds of various curvatures.

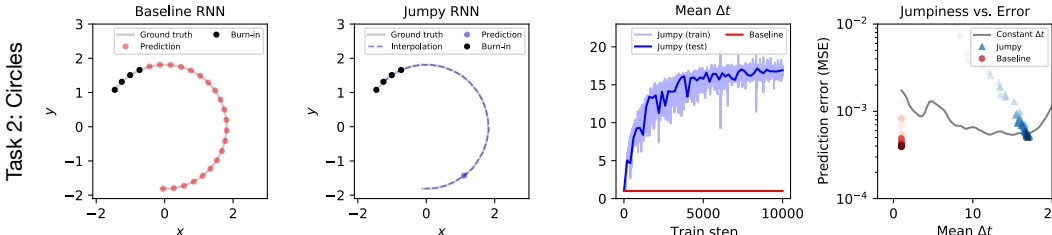

Figure 3: A comparison between a Jumpy RNN and a baseline RNN trained the circles task. Columns 1 and 2 show samples from the models. The Jumpy model summarizes almost the entire trajectory with a single step. Columns 3 shows how jumpiness increases throughout training. Column 4 shows that the jumpy model is able to produce autoregressive samples of comparable quality to the baseline while using twenty times fewer cell updates to do so.

As in the lines task, our model learned to jump over significant spans of time (mean $\Delta_i$ of 17). Figure 3 provides our primary visualization of these results. The first two columns show representative samples from a standard RNN (left) and our Jumpy RNN (right). Dots correspond to RNN updates with black dots corresponding to priming observations. Recall that our model can make a prediction at any time point between updates. We plot these intermediate evaluations with a dashed line and they follow the ground truth trajectory closely. The third column shows the average optimal jump width $\Delta_i^*$ during training. As we can see, it rises over the course of training as the model learns.

The fourth column presents a key summary plot of our experiment – a comparison of mean squared error vs. efficiency in terms of jumpiness. In this chart, further lower-right is better – corresponding to low error with large jumps. In red, we show the progression of error for the standard RNN over the course of training (light to dark). Likewise, we show our model's in blue. Unlike the standard RNN which is at a fixed $\Delta$ (and thus a vertical line in this plot), our approach changes both in error and jumpiness. We see that our approach is significantly jumpier while maintaining a similar error rate.

The gray curve represents the performance of the jumpy model when forced to take constant jumps of size $\Delta$. In this task, the jumpy model converges to a point only slightly beyond this curve, indicating limited utility in dynamic $\Delta_i$ prediction compared to the optimal fixed time step. This is perhaps unsurprising in this case because the dynamics are uniform throughout each sequence. However, our model still was able to automatically identify this point without manual search. As we show in the following section, dynamically predicting $\Delta_i$ can be impactful in more complex settings.

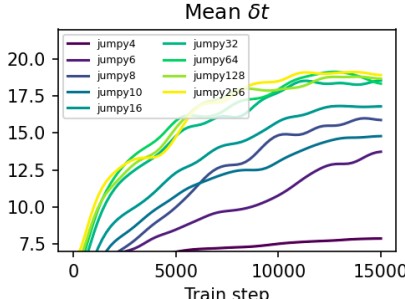

Figure 4: Jumpiness increases as we increase model capacity (hidden state size). We observed this effect across all datasets and experiments.

Finally, we vary the dimensionality of our model's hidden state and retrain. In figure 4, we find a positive relationship between model capacity and jumpiness.

## 3.2 MODELING INTERACTION DYNAMICS

To examine more complex dynamics, we consider a billiards-like simulation in which two balls are bouncing around within a walled enclosure. We examine this setting for 1- and 2-dimensional versions from both coordinate and image-based inputs. Ball trajectories in this setting exhibit periods of essentially linear motion punctuated by collisions. As such, we expect our Jumpy RNN model to adaptively adjust its step size accordingly.

**Datasets.** In each setting, we generate $10,000$ trajectories of 45 time steps each. The environment lacks friction, collisions are perfectly elastic, and walls are placed along 0 and 1 for each coordinate. For coordinate settings, we record trajectories as $x, y$ coordinates for each ball (or just $y$ in the 1D case). For image settings, we render $28 \times 28$ grayscale images depicting a top-down view of the balls.

**Billiards From Coordinates.** The first two rows of Figure 5 visualize the result of 1D and 2D billiards respectively. To visualize samples in the 1D case, we treat time as the x-axis. For 2D, we

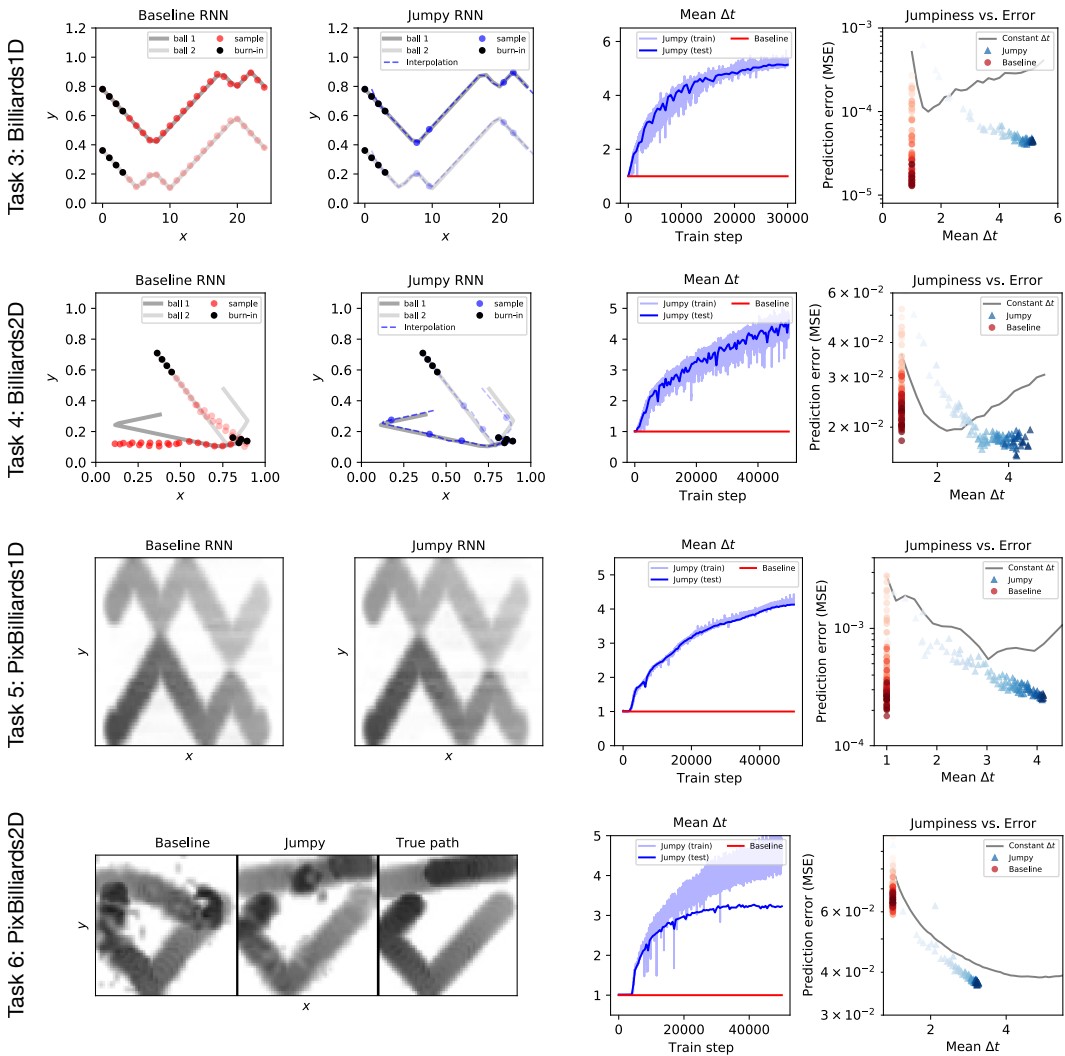

Figure 5: Analyzing a jumpy RNN and a baseline RNN trained on four billiards tasks. Columns 1 and 2 illustrate qualitatively that the jumpy model's predictions were at least as good as the baseline's. Columns 3 and 4 illustrate that our jumpy model's jumpiness varied with data modality (less jumpy behavior on the more difficult pixels task), but that it matched the baseline model's performance while being more efficient to sample from. Forcing the jumpy model to use a constant $\Delta$ gives much worse performance; indicating the learned *adaptive stepping* skips over the most predictable time spans and concentrates on the less predictable ones, like collisions. Note: Supplementary Materials also contains videos for comparing model predictions.

fade from light to dark to indicate time. In samples from both cases, we can see the model learns to concentrate cell updates near collisions. However, we do see instances where collisions are skipped entirely and filled in with linear latent interpolation alone as in the 1D example. The Jumpiness vs. Error plots (last columns) also reflects the impact of this adaptive jumping. Our approach is able to significantly improve in both accuracy and jumpiness compared to using a fixed step size (darkest blue triangle vs. gray curve). For 1D, the baseline model achieves lower error but requires 6x the compute. In the 2D case, our model outperforms the baseline while executing 4x fewer RNN updates.

**Billiards From Pixels.** Learning from pixels directly is a challenging setting. While the underlying motion of the balls is the same as before, the way pixels change as the balls move is extremely non-linear – turning on or off abruptly. For these experiments, we increase the hidden layer size to 512 units and train for 50,000 steps. The third and forth rows of Figure 5 show result summaries for 1D and 2D pixel-based billiards. In the 1D setting, we find that both the baseline and our model

| Task | MSE Scale | Train MSE | | Test MSE | | Sample MSE | | Mean $\Delta_i$ | |
|---|---|---|---|---|---|---|---|---|---|
| | | RNN | Jumpy | RNN | Jumpy | RNN | Jumpy | RNN | Jumpy |
| 1: Lines | $\times 10^{-8}$ | 1.68 | 39.9 | 1.77 | 44.2 | **13.9** | 21.9 | 1 | **20** |
| 2: Circles | $\times 10^{-4}$ | 0.03 | 3.66 | 0.03 | 3.64 | **3.98** | 5.25 | 1 | **17** |
| 3: Billiards1D | $\times 10^{-5}$ | 2.40 | 2.86 | 2.44 | 2.93 | **1.77** | 4.37 | 1 | **5.1** |
| 4: Billiards2D | $\times 10^{-5}$ | 2.66 | 3.55 | 3.09 | 3.73 | 2100 | **1760** | 1 | **4.5** |
| 5: PixBill1D | $\times 10^{-4}$ | 1.26 | 1.49 | 1.56 | 1.74 | **2.37** | 2.64 | 1 | **4.1** |
| 6: PixBill2D | $\times 10^{-3}$ | 1.31 | 3.00 | 2.08 | 3.90 | 64.5 | **37.1** | 1 | **3.2** |
| 7: Weather | $\times 10^{-3}$ | 1.53 / 1.61 | 1.64 | 1.29 / 1.43 | 1.24 | 21.2 / 13.8 | **10.8** | 1 | **2.22** |

Table 1: Summary of train, test, and auto-regressive MSE and mean jump length for all settings. There are two RNN baselines listed in the last row, corresponding to models with $\Delta_i$ fixed to 1 and 2 respectively.

achieve similar error, but our model does so with 4x fewer steps. In the 2D setting however, the baseline RNN yields a 50% higher error rate and generates unrealistic trajectories where balls bend along their trajectory like the sample shown. Reviewing training curves, we noted that the baseline RNN overfits more severely fails to generalize. One hypothesis for our model's success is that the bias towards linear latent dynamics acts as a regularizer in this setting.

**Quantitative Summary.** All experimental settings are summarized with dataset level metrics at convergence in Table 1. We find our Jumpy RNN model performs at or near the same sample error rate while making 3 to 20 times fewer RNN steps. Note that absolute scale of error terms varies by orders of magnitude across tasks and is specified only in the `MSE Scale` column for compactness.

## 3.3 THE JENA WEATHER DATASET

Having determined that the Jumpy RNN could learn useful behaviors on synthetic dynamics tasks of considerable complexity, we were also interested to see how it would perform on real-world data. With this in mind, we trained it on the Jena weather timeseries dataset recorded by the Max Planck Institute for Biogeochemistry (Kolle, 2016). The dataset consists of 14 different features such as air temperature, atmospheric pressure, and humidity. They were collected every ten minutes between the years of 2009 and 2016. This dataset is a common baseline for sequence prediction and forecasting, being used in *Deep Learning with Python* and the TensorFlow time series forecasting tutorial (Abadi et al., 2015; Francois, 2017).

We subsampled the dataset in order to obtain observations every 30 minutes and then divided it into sequences of 80 time steps each. Additional preprocessing steps followed previous work and can be found in Appendix A.3. In addition to training our Jumpy model and a baseline RNN model, we also trained another baseline RNN with a fixed $\Delta_i$ set to 2. Like the gray lines that correspond to fixed $\Delta_i$ in Figure 5, this baseline can be used to infer the advantage of *adaptive* jumpiness over simply subsampling a dataset at a different rate.

We found that our model improved over both baselines in terms of test error and autoregressive sampling error. The difference in sampling error was most significant, nearly halving that of the baseline model. Qualitative results in the Appendix support this finding, with most of the Jumpy RNN predictions being being somewhat better than those of the baseline models. We suspect that part of this improvement in performance comes from the fact that our model performs RNN cell updates less often, and thus does well as sequences increase in length. We also suspect that our model's bias towards learning locally-linear dynamics acted as a regularizer and reduced overfitting to local "inconsequential chaos" (Neitz et al., 2018)

## 3.4 MODEL-BASED PLANNING VIA BACK-PROPAGATION

Our last experiment aimed to leverage the dynamics models we trained in the previous section to measure the value of jumpiness for model-based planning. Specifically, we considered a planning task in the 1D Billiards setting where an agent must set the initial conditions (position and direction) of one ball (ball A) such that the other ball (ball B) reached a fixed goal position ($y = 0.33$) at time $t = 20$. A plan was successful if ball B's final position was within 0.1 of the goal.

We apply a simple planner that starts from a given configuration and iteratively updates ball A's initial conditions by: (1) simulating the forward dynamics via our trained RNN / Jumpy RNN model to estimate the distance between ball B and the goal, and then (2) back-propagating this error through the dynamics model to estimate gradients for the initial conditions and then taking a step in that direction to update the initial conditions.

This procedure exposes interesting characteristics of our approach. Its increased efficiency at predicting the far-future directly impacts planning speed. By taking fewer steps, it may also be less susceptible to exploding or vanishing gradients common to long-horizon sequence models. Even with access to a perfect simulator, this sort of gradient-based planning is not guaranteed to return an optimal plan. However, it does provide a setting for relative comparison.

We evaluate an RNN-Planner and Jumpy-Planner on 1000 episodes, running the planning loop for 20 iterations each. Success rates are shown in Figure 6. While the baseline RNN succeeds slightly more often, it also takes twice as long as our jumpy model ($1.22 \times 10^{-2}$ vs. $6.04 \times 10^{-3}$ seconds / trajectory). This suggests that Jumpy RNNs have potential as efficient dynamics models for planning.

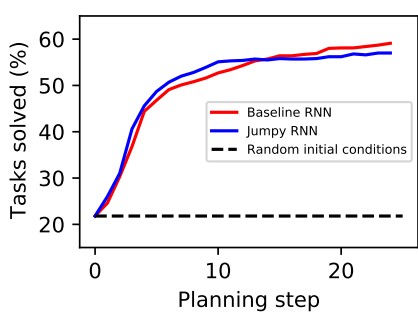

Figure 6: Success rates for RNN- and Jumpy RNN-based planners.

## 4 RELATED WORK

In this section, we review a set of related works that, like our own, aim to model sequences at different timescales or across variable durations. The first and simplest way to model sequences at different timescales is to introduce a multiplicative gating system in order to read, write, and erase memory vectors at different rates (Hochreiter & Schmidhuber, 1997; Chung et al., 2014; Graves et al., 2014). This gating system lets models retain information for longer than they would otherwise, but since the gates are leaky, over time, they still allow noise to dilute the stored information.

**Hierarchical Recurrent Models.** In order to solve the problem of leaky gating, there is a line of work which replaces differentiable gating mechanisms with hard reads, writes, and erases and performs them at different frequencies (Lin et al., 1996; Ling et al., 2015). Relatedly, Koutnik et al. (2014) proposed dividing an RNN internal state into groups and only performing cell updates on the $i^{th}$ group after $2^{i-1}$ time steps. More recent works have aimed to make this hierarchical structure more adaptive, either by data-specific rules (Ling et al., 2015) or by a learning mechanism (Chung et al., 2019).

Skip RNN Campos et al. (2018) is related to Chung et al. (2019) in that it the skip operation is "applied to the whole stack of RNN layers at the same time" thereby shortening the effective size of the computational graph. The Phased LSTM (Neil et al., 2016), meanwhile, adds a learnable periodic time gate to LSTMs so that different neurons can update themselves at different frequencies. More recently, Goyal et al. (2019b) have proposed Recurrent Independent Mechanisms, which consist of a set of recurrent models which make dynamics predictions semi-independently and compete with one another when making a global prediction by way of an attention mechanism. This leads to specialization of individual RIMs and permits some of them to operate at different timescales. We should note that there is a wealth of other works in this general domain including (Voelker et al., 2019; Jernite et al., 2016). A common thread relating these hierarchical recurrent models is that they can all model data at different timescales, but parts of them (the encoders at the very least) all *must* tick at at each time step in a sequence, and hence cannot jump over regions of homogeneous change. Additionally, they are all rest on the notion of discrete "time steps" and do not treat time as a continuous variable.

**Modeling Irregular Timeseries.** There is a parallel lineage of RNN models which, like hierarchical RNNs, permits hidden state updates over variable durations of time. The motivation for these models is to handle sequences with irregular or missing time steps. The simplest approach is to include the time gap between observations as an input to the RNN cell. In practice, though, preprocessing the data by interpolating Che et al. (2018) or masking (Lipton et al., 2016) and then training a regular RNN tends to work better. More recently, several works have proposed applying an exponential

decay function to the RNN hidden state in proportion to the duration of time between observations (Che et al., 2018; Cao et al., 2018; Rajkomar et al., 2018; Mozer et al., 2017). This approach makes hidden state dynamics continuous in time in a limited sense: hidden states are constrained to decay towards zero at a particular rate.

**Neural Ordinary Differential Equations.** One way to learn more complex hidden state dynamics is to predict the derivative of an RNN hidden state and then integrate them over time. That is the core idea of Neural Ordinary Differential Equations, a class of models which has existed since the 1990's (LeCun et al., 1988; Pearlmutter, 1995), but which has recently seen a resurgence of interest (Chen et al., 2018). Recent work suggests that, like exponential decay models, neural ODEs can learn effective hidden state dynamics for irregular timeseries models (Rubanova et al., 2019). Like our jumpy model, these models can be integrated adaptively over time.

One challenge in training neural ODEs is that they must be continuously solved even when no observations occur, making forward passes take 60% - 120% longer than standard RNNs (Rubanova et al., 2019). Since the integration speed of a Neural ODE is proportional to the curvature of the hidden state dynamics (higher curvature requires more steps), one can improve speed and efficiency by regularizing that curvature. Finlay et al. (2020) take a step in this direction. Even so, the number of time steps per function evaluation tends to be less than one. Our model resembles Neural ODEs in that it parameterizes the derivative of a hidden state. However, we make the simplifying assumption that the hidden state dynamics are linear for long stretches of time. This makes our model extremely efficient to integrate over long spans of time – more efficient, in fact, than a baseline RNN. This jumpy behavior is what our model is designed for, and where it excels compared to Neural ODEs.

**Other Related Works.** Recent work by Gregor et al. (2018) in the context of reinforcement learning develops a jumpy planning model which does not use an RNN cell or perform continuous interpolation of latent states. Another relevant work is Embed to Control by Watter et al. (2015) which, like our model, assumes that dynamics is linear in a latent space. As with our work, they train their model on several dynamics problems, some of them from pixels. Unlike our work, their model performs inference over uniform time steps and does not learn a jumpy behavior.

Some related works like Adaptive Skip Intervals by Neitz et al. (2018) and Time-Agnostic Prediction Jayaraman et al. (2019) use a "minimum-over-time" loss across a horizon of future states. This effectively allows them to predict the next predictable state, regardless of how distant in time it might be. In doing so, they decouple dynamics prediction from time and permit temporally-abstract planning. But in doing so, they are unable to reconstruct the dynamics which occur between these predictable frames, they are unable to estimate the "rate of change" of the system, and they do not treat time as a continuous quantity. One way to look at these "minimum-over-time" approaches to temporal abstraction is as a method of finding bottleneck states – states that can be expected to occur with a high degree of certainty McGovern & Barto (2001); Stolle & Precup (2002). Another means of achieving temporal abstraction is to look for "decision states," which occur in situations where the agent diverges from its default behavior in order to "make a goal-dependent decision." Infobot, by Goyal et al. (2019a), investigate this notion of temporal abstraction and propose a model which leverages model-based planning only when it is in such a state. Unlike our model, parts of Infobot still "tick" at every time step and time is not treated as a continuous variable.

## 5 CONCLUSIONS

Having achieved widespread use in commercial and academic settings, RNNs are already a useful tool. But even though they are useful tools, they still have fundamental limitations. In this paper, we reckoned with the fact that they can only forecast the future in discrete, uniform time steps. In order to make RNNs more useful in more contexts, it is essential to find solutions to such restrictions. With this in mind, we proposed a Jumpy RNN model which can skip over long durations of comparatively homogeneous change and focus on pivotal events as the need arises. We hope that this line of work will expand the possible uses of RNNs and make them capable of representing time in a more efficient and flexible manner.

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

## A  APPENDIX

### A.1  EFFICIENT BATCH-WISE LINE SEARCH WITH MASKING.

$$h_i, \dot{h}_i = \text{RNNCell}\left( \left[ h_{i-1}, \dot{h}_{i-1}\Delta_{i-1} \right], \phi\left(\hat{x}(\tau_i)\right) \right) \tag{8}$$

$$\Delta_i = \text{LeakyReLU}(Wh_i + b) + 1 \tag{9}$$

$$h(t) = h_i + (t - \tau_i)\dot{h}_i \quad \text{for} \quad t \in [\tau_i, \tau_i + \Delta_i] \tag{10}$$

In Section 2.2 we defined the optimal jump width, $\Delta_i^*$, and described how to find it for a given step $i$ using the constrained optimization objective in Equation 7. Then we described how one might solve

that equation by starting from $\tau_i$ and iteratively moving through the sample sequence, decoding our model's prediction under the current dynamics, until the loss exceeds $\epsilon$. In this section, we describe how to vectorize this process efficiently over a batch of examples.

At a high level, we will compute the RNN cell update batch at each time step, but only update stored hidden state vectors for instances that exceed the threshold. Let the $H_i$ be a $B \times d$ matrix of hidden state vectors corresponding to a batch of $B$ instances. Likewise for $\dot{H}_i$ and the hidden velocity vectors. Let $\mathcal{M}$ be a $B$ dimensional, binary vector where the $b^{\text{th}}$ entry denotes whether the loss for the $b^{\text{th}}$ instance is greater than $\epsilon$ at the current step. With these definitions, we can write the batched update of $H_t$ via a dynamic masking operation as follows:

$$
H_t = \begin{cases} \hat{H}_t = H_i + (t - \tau_i)\dot{H}_i & \text{if } \ell\left(x_t, \hat{x}(t)\right) < \epsilon \\ \text{RNNCell}\left(\ [H_i,\ \dot{H}_i\Delta_{i-1}],\ \phi\left(\hat{x}(t)\right)\ \right) & \text{otherwise} \end{cases}
$$

$$
= \mathcal{M} \odot \hat{H}_t + (1 - \mathcal{M}) \odot \text{RNNCell}(\cdot) \quad \text{where} \quad \mathcal{M} = \ell\left(x_t, \hat{x}(t)\right) < \epsilon
$$

Since we must compute $\mathcal{M}$ dynamically as we progress over the time dimension, we cannot fuse the RNN cell updates over time, a common technique used to speed up teacher forcing. This, along with the additional cost of decoding $\hat{h}_\tau$ and evaluating prediction error, increases the runtime of a training step of our model by a factor of three over a baseline RNN. Unlike reference RNN implementations, however, our code is not optimized for speed. Furthermore, we are more interested in our model's performance during evaluation than during training, and thus are willing to tolerate this constant slowdown.

## A.2 DISCUSSION OF JUMPY BOOTSTRAPPING

In early experiments, we noticed that our model had a difficult time transitioning from no jumpiness (selecting $\Delta = 1$ at every time step) to some jumpiness (selecting $\Delta = 2$ at rare intervals). Upon further investigation, we found that, having never needed to linearly extrapolate dynamics for more than one time step, our model was very bad at doing so early in training. In fact, the prediction errors at $\Delta = 2$ were consistently much higher than $\epsilon$. We considered two methods of solving this problem. One was to anneal $\epsilon$ starting from a large value to promote jumpy behavior early in training. The other was to allow the model to take a jumpy step 1% of the time, regardless of the value of $\epsilon$. The second option gave the best and most stable results so we used it in our experiments.

## A.3 ADDITIONAL RESULTS FROM JENA WEATHER DATASET

We preprocessed the data as recommended by previous works: first we added $sin$ transforms of the time feature corresponding to day and year frequencies. Then we normalized all features by subtracting the mean and dividing by the standard deviation along each feature channel. The only case in which our methods diverge from previous approaches was in the number of features used: instead of using all 16 features, we used just five: day and year time features, temperature, pressure, and density ($\rho$). During training, we used the same hyperparameters as described in previous experiments except: we used hidden states of size 256, we added weight decay of $1 \times 10^5$. The additional weight decay was necessary because this dataset was smaller than previous datasets and overfitting was a significant problem.

## A.4 VISUALIZING THE LINES TASK

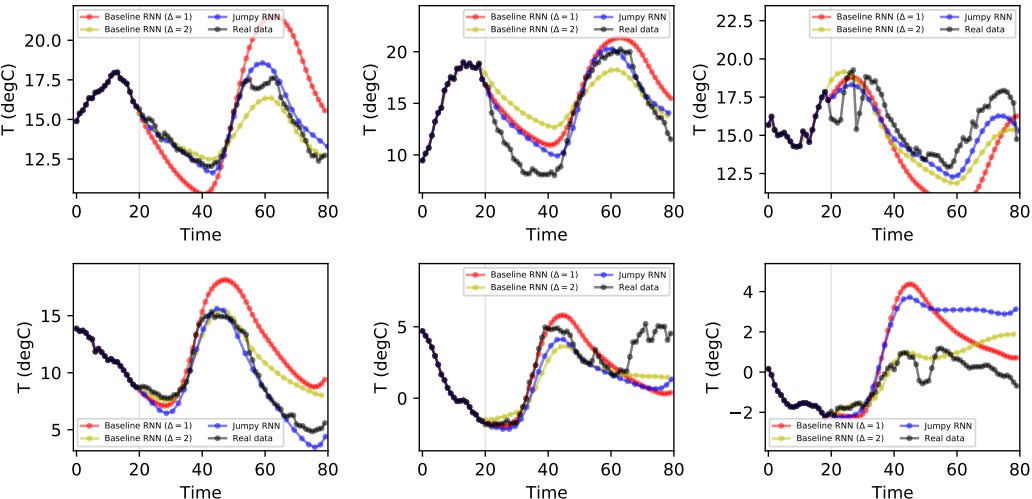

Figure 7: A comparison between a jumpy RNN and two baseline RNNs trained on the Jena weather dataset. Each subplot shows a different test set trajectory and the corresponding predictions of the models. They warmup period occurs during the 20 time steps preceding the vertical gray lines. We see that the Jumpy model consistently produces trajectories that match or improve upon either of the baseline models. We see less variability in jump sizes compared to the synthetic tasks; this is because the inherent noise in the data prevented our model from ever taking jumps larger than $\Delta_i = 4$. In future work, we hope to investigate alternative criteria for jumpiness, such as *cumulative* prediction error, which could mitigate this effect.

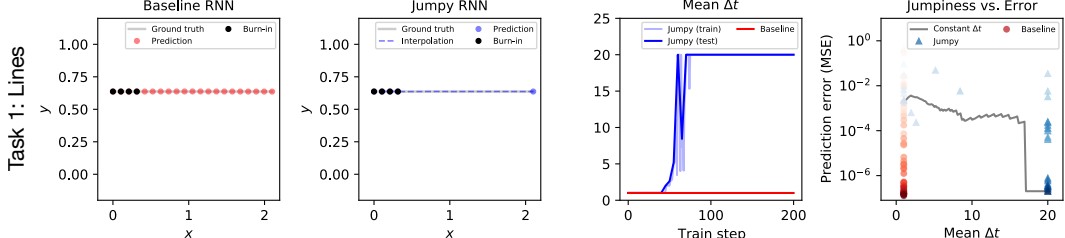

Figure 8: A comparison between a jumpy RNN and a baseline RNN trained on lines task. Columns 1 and 2 show samples from the models. Here we see that the Jumpy model can summarize an entire trajectory with a single linear function, whereas the baseline model cannot. Columns 3 and 4 report key quantitative statistics of the two models. In particular, Column 3 shows how jumpiness increases throughout training. Column 4 shows that the Jumpy model is able to produce autoregressive samples of comparable quality to the baseline while using twenty times fewer cell updates to do so.

