# OpenReview forum: "Jumpy Recurrent Neural Networks"
_ICLR.cc/2021/Conference — Reject_

### Official Review · AnonReviewer3 · 2020-10-19
**Interesting problem but lacking comparisons and experimentation**

**Rating:** 5
**Confidence:** 4

**Review:**

Summary: This work presents Jumpy RNNs, a recurrent network that learns to take variable length steps based on time-scales of the data. The core idea of the paper is to learn a hidden velocity and time span, along with the standard hidden state. The hidden velocity is then used to linearly interpolate the hidden state within the learned time span.  This leads the proposed model to update the hidden state only at the end of time span thus allowing the model to jump over by certain steps proportional to time span. The model has flexibility to produce fine-grained or continuous-time predictions as well as predicting far into the future. Since, Jumpy RNNs do not update the hidden state at every time step, they are computationally efficient than standard RNNs.

Positives:
1. The idea of building a model to provide short term and long term prediction using just one hidden state update is interesting. The problem addressed is important in time series predictions in many domains.
2. The paper is well written and easy to follow.
3. Experiments show that the model works well on simulated datasets.

Concerns:
1. The key concern about the paper is the lack of rigorous experimentation to study the usefulness of the proposed method. The current experiments are mostly based on simulations and are too simple to evaluate the effectiveness of the proposed model. In almost all the cases, the rate of change is constant with few sharp changes. In particular, it would be interesting to see how the model performs on real world datasets, where the hidden state dynamics is unknown, for short-term and long-term time series predictions.
2. Despite the paper stating that there have been some recent work on Neural ODEs (Chen et al., 2018), the paper does not compare with them. Neural ODEs are one of the possible candidates for solving the given problem and it would be interesting to see how fast/slow they are compared to the proposed model.
3. The linear-dynamics of hidden state is too simple and can only handle constant hidden state dynamics or the one with constant slope. Although the authors mention that the hidden-state linearity does not translate to linearity in output space as the learned decoder can be an arbitrarily complex non-linear function, but the experiments are only based on constant or linear hidden state/output dynamics.
4. What's the difference between Test MSE and Sample MSE? Based on the results, standard RNNs achieve lower MSE scores than the proposed model in most cases but use more computations. An interesting comparison would be to compare with a standard RNN with similar computational power as the final learned Jumpy RNN. Possible baseline could be training standard RNN with reduced sampling rate and do linear interpolations in between the predictions.

Other comments:
1. What does baseline model refer to here: 'setting $\epsilon$ to the final training loss of the baseline model'?
2. In the 3rd line of 2nd para in section 3.1, shouldn't it be 'c' instead of 'x'?
3. Could the authors comment on the possibility of using the proposed approach to learn from irregularly sampled time series?

==========Post-rebuttal comments================

Based on other reviews and authors response, I have decided to keep my score. I still feel this paper need more work such as experiments on real world datasets and more comparisons as pointed out in the review.

---

> ### Author Response · Authors · 2020-11-25
> **Answers to questions and additional experimental results. Thanks for your review.**
>
> **[1] “The key concern about the paper is the lack of rigorous experimentation”**
>
> We disagree with the characterization of our experiments lacking rigor, but agree they are all performed on synthetic settings. We deliberately began with simple tasks so as to perform carefully-controlled experiments that would isolate specific aspects of our model. In our experience, leaping directly to noisy real-world datasets can actually lead to less-rigorous conclusions because the causal factors behind a model’s success or failure are more difficult to disentangle from the dataset. The intent of this work was not to obtain the best results on a particular large-scale baseline, but rather to propose and demonstrate a qualitatively new approach to jumpy autoregressive sampling.
>
> _In order to address this concern, we have added a seventh experiment in which we train our models on a real-world weather forecasting dataset._ Our jumpy model performs well against two baseline RNN models; please see the blue diffs in the updated pdf for more details.
>
> **[2] “The linear-dynamics of hidden state is too simple and can only handle constant hidden state dynamics or the one with constant slope.”**
>
> We are not sure what supports the claim that piecewise linear hidden dynamics are too simple. We are also uncertain how simplicity is measured or what the claim suggests they are “too simple” for. Piecewise linear approximations are arbitrarily accurate as the span of each linear segment decreases and our model has control of this span via predicted Deltas. As the reviewer notes, our piecewise linear dynamics are in latent space and are further passed through arbitrarily complex decoders that are learned end-to-end with the linear hidden dynamics.
>
> **[3] “experiments are only based on constant or linear hidden state/output dynamics.”**
>
> This is not exactly true. It is true that our settings have simple linear motion; however, collisions are nonlinear events and occur frequently. Moreover, the pixel settings (Task 5 and 6) certainly lack linear input and output dynamics when encoding and generating pixel grids.
>
> **[4] “What's the difference between Test MSE and Sample MSE?”**
>
> Test MSE is the test loss of our model; it is being trained using teacher forcing and a MSE loss function over predictions. Sample MSE is the same metric, computed between a test trajectory and an autoregressive sample from our model, after our model was run on the first four observations of the test trajectory. This is called “Autoregressive error” in some works.
>
> **[5] “Compare with a standard RNN with similar computational power” or “with reduced sampling rate and do linear interpolations in between the predictions.”**
>
> Our jumpy model has a complexity comparable to the standard RNN models. If anything, its representational capacity is diminished because only half of the cell state $h$ is used to represent state whereas the other half is used to represent velocity.
>
> Running a standard RNN at a range of step sizes and then selecting the best would be unfair to our model -- essentially assuming some oracle could tell us the optimal step size a priori. When in practice, it can be quite difficult to know what step size would be best for a given dataset. Our approach discovers this dynamically during training. Further, running an RNN at a higher step rate produces no output for steps that have been skipped. As the reviewer suggests, we could perhaps resolve this by interpolating in the coordinate-based tasks but it is unclear what to do when the output is pixels. In either case, this would constitute an additional method wrapping around a standard RNN which could itself be a research topic.
>
> **[6] Comparison to Neural ODEs**
>
> While both Neural ODEs and Jumpy RNNs define continuous time dynamics, they are composable techniques.
>
> Neural ODEs are dynamics functions whereas Jumpy RNNs are models that predict dynamic functions. You could image, for example, a Jumpy RNN that predicts a sequence of Neural ODEs with each being integrated over the range Delta. Given this composability, we did not compare directly, but do agree comparing these approaches purely as continuous models could be valuable.
>
> **[7] Response to minor comments**
>
> Q: “What is the baseline model in ‘setting to the final training loss of the baseline model?’”
> A: In that context, the “baseline model” refers to the regular RNN model with a GRU cell that ticks through time at a constant rate. We have clarified this point in the paper.
>
> Q: “Shouldn’t it be ‘c’ instead of ‘x’?”
> A: Yes, that is correct. We have made the correction in the paper; thanks for pointing this out
>
> Q: “...possibility of …[learning] from irregularly sampled time series?”
> A: This is certainly possible, and we have preliminary experiments to this effect. In order to do this, one would simply set \Delta t to the interval (in time) between adjacent observations. One could use the same criterion used in Equation (7) to train such a model.

---

### Official Review · AnonReviewer4 · 2020-10-27
**Unconvincing experiments and lack of novelty**

**Rating:** 5
**Confidence:** 4

**Review:**

This paper proposes Jumpy Recurrent Neural Network, an RNN model with non-uniform time steps. To train this model, the authors propose to use a greedy supervision to determine optimal time intervals. The experiments on linear dynamics prediction and planning show comparable performance of proposed model against standard RNNs. The advantage is that the proposed model can significantly speed up RNNs under relatively linear dynamics prediction tasks. Even though non-uniform time step RNNs have been studied in many literature, this proposed training supervision method seems novel.

+ves:

+ Overall, the paper is well written. In particular, the proposed model and its training methods are clearly explained.

+ The results section is well structured. The hyperparameters, architecture, and experimental settings are well demonstrated.


Concerns:

- The proposed method, non-uniform time interval RNN, is not novel. For example, Che et al 2018, Lipton et al 2016, have very similar contributions. This paper needs to clarify the difference.


- The proposed greedy training method seems have significant flaw. Clearly it will have Delta=1 at the beginning of the training. Then it can only improve corresponding Delta one by one at each timestep. The loss corresponding to Delta is highly non-continuous and it will definitely not learn to find the real optimal Delta.


- The model does not seem to learn meaningful Delta. See Figure 3 and 4. For such clean data, the model should learn to predict only turning points but it seems it just predict random intervals. Also, there are no verification of learning meaningful Delta.


- The experiment is clearly not fair for a standard RNN. Since Delta=1 is always true for standard RNN, clearly it will be slower. The speedup ratio will only be the average of predicted Delta in the proposed model. However, what if the RNN just sets a larger Delta, what if the RNN just set a random predicted Delta?


- The last experiment is not convincing. First, it's a simple supervised task, not real "model-based planning". Second, Figure 6 seems not converged. No experimental detail is given for this task. Third, twice as fast mean the model generally just predict Delta=2.


- All the experiments have internal bias towards linear dynamics. This favors the model with corresponding inductive bias. To prove the model is useful, I would recommend to run standard RNN synthetic task like copying task, adding task etc, to prove its basic functionality.


Minor comments:

* Figure 1, in basic RNN, it should be an arrow instead of a line from h_i to GRU.

* Equation 3, people usually don't write output as x, otherwise it can only be a sequence prediction task like language modeling. Please write it y. Also, it creates confusion to equation 7 whether it's input or output.

* The "Jump Bootstrap" seems very important technique for the proposed method. But there's no experiment or discussion on it.

=====POST-REBUTTAL COMMENTS========

I thank the detailed response from the authors. The authors addressed the novelty of this paper. The experimental results on toy tasks are convincing. However, the way this method increases Delta still seems very problematic to me and not seem robust in complex real world cases.

I increased my score.

---

> ### Author Response · Authors · 2020-11-25
> **Part II: Response to concerns 4, 5, and 6**
>
> **[4] “The experiment is clearly not fair for a standard RNN…what if the RNN just sets a larger Delta?”**
>
> We disagree. For both models, a single training run is performed starting from the data’s native sample rate -- a reasonably standard thing to do for sequence modelling. Running a standard RNN at a range of step sizes and then selecting the best would make it unfair to our model -- essentially assuming some oracle could tell us the optimal step size a priori. When in practice, it can be quite difficult to know what step size would be best for a given dataset. Our approach discovers this dynamically during training.
>
> It’s worth noting that running an RNN at a higher step rate produces no output for steps that have been skipped. We could perhaps resolve this by interpolating in the coordinate-based tasks but it is unclear what to do when the output is pixels. In either case, this would constitute an additional method wrapping around a standard RNN. In contrast, our approach can output predictions at any point in time, regardless of step rate.
>
> **[5] “The last experiment is not convincing.” It is (a) “a simple supervised task, not real model-based planning”, (b) “Figure 6 seems not converged.”, and (c) “the model generally just predict Delta=2”**
>
> We object to all three assertions:
> * (a) This is certainly a form of model-based planning. We learned a dynamics model and then leveraged rollouts from that model to make a plan for a finite horizon, which was then executed in the environment. This is similar to continuous trajectory optimization ubiquitous in robotics. Further, Reviewer 1 points out that this style of planning was also explored in related works and suggests pushing further in this direction. It is unclear to us from the reviewer’s comments what precisely they take exception to here or why it is a “simple supervised task” -- hopefully we can clear this misunderstanding.
> * (b) We ran this experiment for longer and across several seeds; neither model exceeded 60% performance on the task and we believe that the plot is a fair representation of asymptotic performance. Section 3.3 gives the experimental setup for this task -- what experimental details do you see missing? Note that some experimental details were fixed across all experiments -- these can be found at the beginning of Section 3. Let us know if you feel something could be clearer.
> * (c) This is not true because the individual jumps taken by our model vary widely: even though the average jump length is 2, individual jumps are frequently much smaller or much larger.
>
> **[6a] “All the experiments have internal bias towards linear dynamics.”**
>
> This is true, but ignores important details. Our model’s bias is towards linear dynamics in the latent space, not the output or observation spaces. This is why we can handle highly non-linear input/outputs like the pixel billiards task. That said, we view this internal bias towards linear dynamics as a desirable property of our model. Locally-linear approximations of dynamics can be highly effective and many physical processes are linear for long spans of time in the right state space. _The same critique could be posed against the linear bias of Kalman filters, yet they remain a useful and widely used tool._
>
> **[6b] run standard RNN synthetic tasks like copying, adding, etc, to prove its basic functionality.**
>
> _We note that our focus is on continuous-time sequences._ Discrete symbol processing tasks like the reviewer suggests are technically possible, but are non-standard as a benchmark for continuous models. In the usual form of the suggested tasks, each symbol is not predictive of the next so our model would revert to a Delta of 1 (achieving high error any time longer steps were attempted). _At that point, the jumpy model would be roughly equivalent to a standard RNN and perform similarly._ Alternative forms could be derived where long stretches of useless symbols are inserted after identifiable markers, but this is not our focus and beyond the scope of this work.

---

> ### Author Response · Authors · 2020-11-25
> **Part I: Response to concerns 1, 2, and 3**
>
> We appreciate the reviewer’s positive comments about our writing and the quality of our experimental results. We believe there are some misunderstandings about our work and its context and we respond to the reviewer’s first three concerns below. We respond to the next three in a second comment.
>
> **[1] “The proposed method, non-uniform time interval RNN, is not novel.”**
>
> We strongly object to the notion that our approach is not novel. To our knowledge, our work is the first to define a continuous latent state as a series of predicted linear-latent functions and their associated time spans. There is of course a rich history of work on temporal abstraction, including RNN-type sequence models that operate at non-uniform intervals. _We cite and discuss both Che et al (2018) and Lipton et al (2016) along with other relevant works in the related work section._ For convenience, we reiterate that discussion below.
>
> Che et al (2018) define a continuous time RNN state by applying exponential decay to the hidden state between sequential, fixed time-steps in order to deal with irregularly sampled input data. No notion of jumpiness is proposed and the model operates otherwise as a standard RNN. In contrast, our jumpy model parameterizes the hidden state between ticks as a predicted linear function over a predicted, dynamic span of time. Similarly, Lipton et al (2016) study the use of indicator variables for missing data whereas our model “learns to take variable length steps based on time-scales of the data.”. Critically, neither work examines how to perform autoregressive sampling with non-uniform tick rates [see R4], we do. In response to Reviewer 1, we have extended our Related Work section to add further context to similar works.
>
>
> **[2] “The proposed greedy training method seems to have a significant flaw...Then it can only improve corresponding Delta one by one at each timestep.”**
>
> This is a misunderstanding of our approach -- Deltas can change by arbitrary amounts. The procedure described in Section 2.2 solves for the maximum jump such that prediction loss over the jump is less than an error threshold Epsilon (Eq. 7). As error is monotonic, this maximization is exact. In other words, if a model’s linear latent state function correctly predicted the entire sequence, Delta would be set to that duration regardless of its previous value.
>
> **[3] “The model does not seem to learn meaningful Delta[s].”**
>
> We disagree. From the comment, we take the reviewer to have equated “meaningfulness” of Deltas to their correspondence to collisions. This is one interpretation and looking at Task 3 (Figure 5), _we see all jump points in this example correspond closely to collisions in one (or both) of the lines_. Perhaps the reviewer did not realize the model is joint over the pair of lines and thus ticks whenever either collides in this random example. Or maybe mistook the four warm-up inputs as short jumps. We will clarify. For more complex task settings, this alignment of ticks to collisions is indeed less coherent.
>
> However, Deltas functionally correspond to jump lengths of bounded error in training. As a result, what is easy / difficult to model (and thus where the model ticks) depends on the complexity of the task and the model capacity. For example in Figure 4, increasing the capacity of our model in the circle task results in increased jump lengths, eventually saturating at the average full arc length.
>
> Further, our results show that predicted Deltas are dynamic, context-dependent, and lead to “meaningful” improvements over a jumpy RNN operating at a static tick rate. In our plots in Figure 5, 4th column -- we show a gray line corresponding to running the trained jumpy model at fixed Deltas. Across all datasets we find improved performance for the full model that predicts Deltas dynamically.

---

### Official Review · AnonReviewer2 · 2020-10-28
**Well-written paper with simple and interesting ideas**

**Rating:** 7
**Confidence:** 4

**Review:**

## Paper Summary

This paper proposes a recurrent network architecture for future prediction where the hidden states (and the outputs) aren't updated step by step as done traditionally. Instead, the network models the hidden state dynamics as being piecewise linear over varying time spans. It learns to produce the linear dynamics together with each time span, and can "jump" to the next time span according to its own predictions. Hidden states for any time step within a span can be easily obtained by interpolation using the predicted linear dynamics. Experiments on a series of synthetic benchmarks are used to demonstrate that the model can learn to utilize this structure to reduce the amount of computation.

## Strengths

- The proposed approach is simple and intuitive. The study is likely to be valuable to researchers in sequence modeling, by showing that these simple ideas can work surprisingly well. One might expect that the non-stationarity of the "jump size" targets might make training such models too difficult, but the authors show that other than requiring a couple of tricks (setting $\epsilon$ using a baseline and "jump bootstrapping"), training works well.

- The paper is an enjoyable read due to its clarity and presentation. The claims made are modest and clear without over-reaching, the experiments use tasks that are directly motivated by the goals of the study, and the related work is clearly and fairly discussed.

## Weaknesses

- A clear weakness of the paper is that all experiments were conducted on synthetic tasks of low complexity. Although these experiments were illustrative and helpful, it is unclear if the approach works reasonably well on more realistic problems. One "failure" mode could be that the Jumpy RNN tends to jump almost every step for noisy and real-world data. At this stage this study does not really say whether this architecture is also likely to work well on such problems, but is more of a proof of concept.

## Review Summary

I think the strengths of this paper outweigh the weaknesses. While the experiments were not conducted on common benchmarks or real-world data, I think this paper is well-written as a proof of concept, does not overclaim and should motivate further work on similar ideas. My recommendation is to accept.

## Questions for Authors

1. Have you tried more complex datasets or RL problems? Are there any negative results you can report?
2. Have you found the non-stationarity of the targets to be issue for training under any conditions?

## Post-rebuttal

I thank the authors for their hard work, and for incorporating my suggestions into the paper. I believe the paper has improved.

---

> ### Author Response · Authors · 2020-11-25
> **Answers to your questions as well as additional experiments. Thanks for your review**
>
> Thanks for your very clear summary of the paper, positive comments about the technique and its usefulness to the community, and constructive questions. We reply below.
>
> **[1] “clear weakness of the paper is that all experiments were conducted on synthetic tasks of low complexity”**
>
> _We have added an experiment featuring a real-world dataset in response to your request._ It is the Jena weather dataset which is used as a canonical real-world time series forecasting baseline in a number of previous works. We show that our model matches and, in the case of autoregressive sample error, improves over two baseline RNN models, while using fewer steps to make these predictions. Please refer to the new Section 3.3 (in blue) in the updated pdf for details. We hope that this addresses your comment!
>
> **[2] “Have you tried more complex datasets or RL problems? Are there any negative results you can report?”**
>
> We have converted the billiards 2D environment into an RL environment and begun using it to benchmark our two models. We found that adding action information (representing the forces applied to the cue ball) was difficult to integrate into both the baseline and the Jumpy model, especially when these actions are taken at a sparse rate (eg every 30 timesteps, on average). We plan to address these challenges in the future, but it is beyond the scope of this work.
>
> **[2.3] “Have you found the non-stationarity of the targets to be [an] issue for training under any conditions?”**
>
> We have not encountered significant trouble from the non-stationarity of the $\Delta$ targets. In fact, early experiments that kept $\Delta$ fixed for multiple epochs (i.e.more stationary targets) yielded slower convergence and worse results. Early in training (the first thousand steps of gradient descent) the training loss decreases a bit more slowly as the $\Delta t$ predictor adjusts to changes in the representation of the internal state. But aside from these discrepancies very early in training, we do not observe adverse effects.
>
> _Thanks again for your positive feedback and constructive criticism. We believe it has improved the paper._

---

### Official Review · AnonReviewer1 · 2020-10-28
**Jumpy RNNs**

**Rating:** 5
**Confidence:** 5

**Review:**

Main problem: The paper tries to tackle the problem of deciding how to make jumpy predictions or more precisely event based updates in a RNN as compared to clock based updated. Normal RNNs/LSTMs update their hidden states at every time step. Predicting the hidden states at each time step often lead to compounding errors i.e., small errors accumulate while making predictions for many time steps in the future. I really like this problem.

Method: The paper proposes a method to update hidden states in an event driven manner. The paper defines the hidden state as a continuous, piece wise linear function. The paper proposes to predict the jump interval as well a hidden velocity which basically signifies the "change" in hidden dynamics over the jump interval, thus the update function of the proposed method is a function of both the jump interval as well as hidden velocity.  In order to encourage the jump_interval of more than 1, the paper defines an auxiliary parameter such that the prediction loss b/w the actual input and predicted input is less than \epsilon, where epsilon is a hyper-parameter.

Strong points:

- I really like the underlying idea, irrespective of the results.
- I like the preliminary results in figure 3 which basically justifies the core behind the proposed method.
- I also like the idea, and preliminary results for jumpy planning. it was also studied in [a, b].

Questions:

1.  A main drawback of the proposed method is the reliance on the \epsilon parameter. In many cases, it would be the case that their are multiple entities in the environment, like for ex 2 bouncing balls as compared to one, and then the model has to adaptively decide as to for which part of the input, the model should skip the update. So, I'm curious as to how the model with perform when their are multiple bouncing balls (or multiple entities in the env.) against each other.

2. It would also be interesting to study the generalization performance of the proposed method. Does the ability to skip update gives the ability to generalize better to out of distribution examples.

3.  "Finally, we vary the dimensionality of our model’s hidden state and retrain. In figure 4, we find a positive
relationship between model capacity and jumpiness" . The paper mention that they find a positive correlation b/w the model capacity and jumpiness.  Does this behaviour exists in all the datasets which the paper explore ?

4. Related work: I think, many of the important references are missing. There are various important references. Like there's some work done where the idea is to learn how to skip updates in RNN like in SkipRNN (c), or in Adaptive Skip Intervals (d), or in the context of RIMs (e), where each module decides whether to update it's hidden state or not update their hidden state. There's also some work where the idea is to learn event based representations like in PhaseLSTM (f). It would be nice to see how the proposed method compares to any of these methods.

References:

- (a) Time-Agnostic Prediction: Predicting Predictable Video Frames, https://arxiv.org/abs/1808.07784
- (b). InfoBot, https://arxiv.org/abs/1901.10902
- (c). Skip RNN: Learning to Skip State Updates in Recurrent Neural Networks, https://arxiv.org/abs/1708.06834
- (d). Adaptive Skip Intervals: Temporal Abstraction for Recurrent Dynamical Models, https://arxiv.org/abs/1808.04768
- (e). Recurrent Independent Mechanisms, https://arxiv.org/abs/1909.10893
- (f). PhaseLSTM, https://papers.nips.cc/paper/6310-phased-lstm-accelerating-recurrent-network-training-for-long-or-event-based-sequences.pdf

5. I'm not sure what's the take away message from the paper as of now. In the current form the paper shows computational benefits i.e., the proposed method can achieve similar results, by updating dynamically based on events, and hence establishes a trade-off between the temporal resolution of the input and the computational expense. The results for planning are also very interesting. So, one suggestion could be to do extensive experiments and formulate the entire paper in the context of planning because as of now, the bouncing ball results for test mse are not better as compared to the baseline.

=======

After Rebuttal: I have read the rebuttal, as well as reviews by other reviewers. I really like the idea, but its important to evaluate the idea with respect to a downstream task to get a better idea on how to use the learned structure.

---

> ### Author Response · Authors · 2020-11-25
> **Part II: Responses to other questions**
>
> **[2] “A main drawback...is the reliance on the \epsilon parameter.  [...] I'm curious as to how the model will perform when there are multiple bouncing balls (or multiple entities in the env.) against each other.”**
>
> We agree that adding a hyperparameter can represent a technical challenge, especially during training. However, in the process of constructing any jumpy dynamics model, one must introduce an additional hyperparameter that controls the tradeoff between length of jump and accuracy. In works (a) and (b) the authors use the hyperparameter $\beta$ to weight the time horizon and information regularization term, respectively. In work (c), the authors use an “update budget” hyperparameter $\lambda$ to control the skip rate. The other works do likewise. Fortunately, we found that our model was stable and resilient to non-optimal values of $\epsilon$. For example, multiplying $\epsilon$ by 0.5 or 2 did not substantially impact our model’s performance.
>
> **[3] “Does this behavior [increasing jump sizes with increasing capacity] exist in all the datasets which the paper explores?.”**
>
> Yes, this behavior occurs in all the datasets we explored. It’s a good question. We have noted this in the caption of Figure 4.
>
> **[4] Does the ability to skip updates give the ability to generalize better to out of distribution examples?**
>
> We did not examine the OOD setting explicitly; however, as the environments became more complex (e.g. 2D billiards) the training set represented a sparser sampling of possible dynamics. Our jumpy model generalized better than the baseline RNN. Ball paths generated by the baseline often veered left or right in the absence of external forces, producing non-physical dynamics. Our jumpy model achieved lower sample error, and when it did make mistakes, they were often related to the fact that a jumpy forward prediction overshot a collision.

---

> ### Author Response · Authors · 2020-11-25
> **Part I: Discussion of references and corresponding changes to the paper**
>
> We appreciate the reviewer’s positive comments and their thorough comments. We respond to these below.
>
> **[1]  “Related work: I think, many of the important references are missing”**
>
> Thank you for your recommendations regarding related work. _We have added all six of them to the paper, along with extensive discussion (see blue paragraphs in the new pdf)._ This area of research is quite rich, and we want to do our best to cover it thoroughly. Broadly, we see related works as falling into two categories: Works which are truly jumpy in time, i.e. skipping arbitrary lengths of time without expending computation based on the duration (a,b,d); and works which focus on hidden state update mechanics that enable multi-time-scale reasoning (c,e,f). In addition to the updates to the paper, here is some detailed commentary on the six references:
>
> --- In work (a), _Time-Agnostic Prediction...’_ the authors perform jumpy planning by predicting bottleneck states in a time-agnostic manner (using a minimum-over-time loss). The key differences between their approach and ours is that 1) their model cannot interpolate between bottlenecks in order to reconstruct the intervening dynamics and 2) their model is a conditional VAE or a conditional GAN, and does not maintain a hidden representation of the state of the system.
>
> --- In work (b), _Infobot:…_ the authors leverage model-based planning information only at decision states; elsewhere they use “...model-free knowledge to navigate between bottlenecks”. The key difference between our approach and theirs is that our model is designed for continuous-time, model-based planning whereas theirs is built upon an underlying notion of discrete states and does not perform jumpy computation (though it does run model-based planning adaptively).
>
> --- In work (c), _Skip RNN..._ the authors propose a model similar to Hierarchical Multiscale RNNs (HM-RNNs) except the skip operation is “applied to the whole stack of RNN layers at the same time.” This work differs from ours in the same way that HM-RNNs differ: even though some model updates can be skipped during the forward pass, there are still parts of the model that need to “tick” at every step (in particular, the encoder). In contrast, our model jumps directly over such intervals. The difference is most pronounced in the context of planning, where the Skip RNN model would need to sample N time steps in N ticks, whereas ours can simulate the same number of time steps in many fewer ticks.
>
> --- In work (d), _Adaptive Skip Intervals:..._ the authors propose a model which performs temporally-abstract forward simulation in a manner similar to (a): they use a minimum-over-time loss to predict the next predictable state. In their experiments, this corresponds to, for example, predicting collisions with ramps in a funnel board domain. Like our work, theirs is able to perform jumpy forward sampling of dynamics. Unlike our work, theirs does not match forward prediction to physical time steps and cannot reconstruct the dynamics that occur during a jumpy transition.
>
> --- In work (e), _Recurrent Independent Mechanisms_ the authors propose a recurrent model which consists of a set of recurrent models which make dynamics predictions semi-independently and compete with one another when making a global prediction by way of an attention mechanism. This leads to specialization of individual RIMs and permits some to function at different timescales (eg Figure 2). This work is analogous to Hierarchical Multiscale RNNs in that, during sampling, parts of this model will still need to “tick” at every time step (the encoder will, at the very least). Our model differs in that it permits continuous-time dynamics and does not need to perform any computation/”ticks” at a given time step.
>
> --- In work (f), _Phased LSTM:..._ the authors propose a learnable periodic gating function for an LSTM that enables different neurons to update themselves at different rates. As a result, “it acts like a learnable, gated Fourier transform on its input” and this is useful in a number of cases. This work is closely related to Koutnik et al (2014), which we discuss in Related Work, except in this work the periodic gating function is learnable. Like our model, this work permits computation across adaptive timescales; unlike our model, it does not learn continuous-time dynamics and (parts of) it _must_ “tick” at every time step in a sequence.
>
> --- We have also added a few other related works including: [Legendre Memory Units](https://bit.ly/2KEWGLO),  [Variable computation in RNNs](https://arxiv.org/abs/1611.06188), and two papers about bottleneck discovery in RL (McGovern & Barto, “Automatic discovery...” and Stolle & Precup “Learning options...”)
>
> _In addition to related work, the author brought up several other questions. We respond to these in the next comment "Part II: Responses to other questions."_

---

### Decision · Program_Chairs · 2021-01-07
**Final Decision**

**Decision:**

Reject

**Comment:**

The authors propose a "jumpy RNN" to adaptively change the step size of an RNN to match the time scales of the system dynamics. Reviewers found merit in the simple and intuitive idea, but were less enthusiastic about the experimental results and the comparison to existing work. (Adaptive step size methods have been a subject of recent work in RNNs, not to mention in numerical methods for ODE solvers.) Overall, I think the additions the authors made in the discussion phase did strengthen the paper, but further work is necessary before publication.